# Glutathione-S-Transferase Theta 2 (GSTT2) Modulates the Response to Bacillus Calmette–Guérin Immunotherapy in Bladder Cancer Patients

**DOI:** 10.3390/ijms25168947

**Published:** 2024-08-16

**Authors:** Juwita N. Rahmat, Sin Mun Tham, Ting Li Ong, Yew Koon Lim, Mugdha Vijay Patwardhan, Lata Raman Nee Mani, Revathi Kamaraj, Yiong Huak Chan, Tsung Wen Chong, Edmund Chiong, Kesavan Esuvaranathan, Ratha Mahendran

**Affiliations:** 1Department of Surgery, Yong Loo Lin School of Medicine, National University of Singapore, Singapore 119228, Singapore; juwita.r@nus.edu.sg (J.N.R.); surtsm@nus.edu.sg (S.M.T.); surlimy@nus.edu.sg (Y.K.L.); mugdha.p@u.nus.edu (M.V.P.); lraaman@gmail.com (L.R.N.M.); surpr@nus.edu.sg (R.K.); suresuva@nus.edu.sg (K.E.); 2School of Engineering, Biomedical Engineering, Temasek Polytechnic, Singapore 529757, Singapore; 3Biostatistics Unit, Yong Loo Lin School of Medicine, National University of Singapore, Singapore 117597, Singapore; medcyh@nus.edu.sg; 4Department of Urology, Singapore General Hospital, Singapore 169608, Singapore; chong.tsung.wen@singhealth.com.sg; 5Division of Surgery & Surgical Oncology, National Cancer Center Singapore, Singapore 168583, Singapore; 6Department of Urology, National University Hospital, National University Health System, Singapore 119074, Singapore

**Keywords:** glutathione S-transferase theta 2, polymorphism, genetic, BCG vaccine, urinary bladder neoplasms, bladder cancer, immunotherapy

## Abstract

Glutathione-S-transferases (GST) enzymes detoxify xenobiotics and are implicated in response to anticancer therapy. This study evaluated the association of GST theta 1 (GSTT1), GSTT2, and GSTT2B with Mycobacterium bovis Bacillus Calmette–Guérin (BCG) response in non-muscle-invasive bladder cancer treatment. In vitro assessments of GSTT2 knockout (KO) effects were performed using cell lines and dendritic cells (DCs) from GSTT2KO mice. Deletion of GSTT2B, GSTT1, and single-nucleotide polymorphisms in the promoter region of GSTT2 was analysed in patients (*n* = 205) and healthy controls (*n* = 150). Silencing GSTT2 expression in MGH cells (GSTT2B^FL/FL^) resulted in increased BCG survival (*p* < 0.05) and decreased cellular reactive oxygen species. In our population, there are 24.2% with GSTT2B^Del/Del^ and 24.5% with GSTT2B^FL/FL^. With ≤ 8 instillations of BCG therapy (*n* = 51), 12.5% of GSTT2B^Del/Del^ and 53.8% of GSTT2B^FL/FL^ patients had a recurrence (*p* = 0.041). With ≥9 instillations (*n* = 153), the disease recurred in 45.5% of GSTT2B^Del/Del^ and 50% of GSTT2B^FL/FL^. GSTT2^FL/FL^ patients had an increased likelihood of recurrence post-BCG therapy (HR 5.5 [1.87–16.69] *p* < 0.002). DCs from GSTT2KO mice produced three-fold more IL6 than wild-type DCs, indicating a robust inflammatory response. To summarise, GSTT2B^Del/Del^ patients respond better to less BCG therapy and could be candidates for a reduced surveillance regimen.

## 1. Introduction

Non-muscle-invasive bladder cancer (NMIBC) is associated with frequent recurrence, which can be reduced by Mycobacterium bovis Bacillus Calmette–Guérin (BCG) immunotherapy [1]. The EAU guidelines recommend an induction course of six weekly BCG instillations (induction therapy) followed by a repeat of three weekly maintenance instillations every 3 months for up to a year for intermediate risk and up to 3 years for high risk NMIBC [2]. However, 30–50% of patients fail BCG immunotherapy [3], and because of the necessary frequent surveillance, there is a high cost associated with disease management [4]. Given the worldwide shortage of BCG [5], predictive markers for therapeutic modulation have become important. Based on the mechanisms of action of BCG that have been elucidated [6], single-nucleotide polymorphisms (SNPs) in genes associated with cytokine production [7], DNA repair pathways [8], and reactive oxygen species production [9,10] have been evaluated for their predictive value in response to BCG therapy.

We previously screened human bladder cancer cell lines for upregulated genes after a two-hour exposure to BCG [11] to identify more predictive markers and found the glutathione-S-transferase theta 2 (GSTT2) gene. GSTT2 is a member of the glutathione-S-transferase (GST) family of proteins [12] that detoxify xenobiotics and endobiotics by their conjugation to glutathione (GSH). The theta family is located on chromosome 22q11.2 and comprises GSTT1, GSTT2, and a pseudogene GSTT2B (a duplicate of GSTT2 that lies in a head-to-head arrangement) [13]. GSTT1 and GSTT2 share 55% homology but have less than 15% homology with other GST proteins. The theta family of proteins detoxifies distinct substrates. GSTT1 acts on dichloromethane and related compounds, whereas GSTT2 acts on cumene hydroperoxide, ethacrynic acid, and 1-menaphthyl sulfate [14].

The GSTT2B pseudogene is often lost in humans [15]. The loss of the pseudogene reduces GSTT2 expression significantly [15]. GSTT2 expression is associated with a reduced risk of colon cancer [16], oesophageal squamous cell carcinoma [17], and Barrett’s oesophagus [18]. Promoter SNPs in GSTT2 have been associated with colon cancer risk in the Korean population [19]. GSTT1 is also frequently deleted in humans, and its loss has been associated with an increased risk of developing bladder cancer [20]. The GSTT1-positive genotype is also an independent predictor of early BCG failure in patients who received an induction course of BCG [21]. In plants and invertebrates, GSTT proteins have been shown to modulate innate immune responses [22,23]. Thus, we hypothesised that GSTT2 modulates either the development of bladder cancer and/or the response to BCG immunotherapy.

In this study, we evaluated the GSTT2B status of a panel of human bladder cancer and promonocytic cell lines. Cell lines with homozygous GSTT2B deleted (GSTT2B^Del/Del^) or full-length (GSTT2B^FL/FL^) genotypes were selected for analyses. To examine the effects of GSTT2 on the response to BCG, GSTT2 was overexpressed in GSTT2B^Del/Del^ cell lines and suppressed in GSTT2B^FL/FL^ cell lines. The impact of GSTT2B and GSTT1 deletions and selected GSTT2 promoter SNPs [18] on the development of bladder cancer and the response of bladder cancer patients to BCG immunotherapy were determined. Finally, the effect of GSTT2 on immune activation in response to BCG was investigated using bone marrow-derived dendritic cells (DCs) from GSTT2 knockout (KO) and wild-type (WT) mice. The primary aim is to characterise the effects of the respective gene variances and polymorphisms on the risk of developing bladder cancer and its association with BCG treatment outcomes. Our results indicated that GSTT2, but not GSTT1, modulated the response to BCG immunotherapy in our population, and neither gene impacted the development of bladder cancer. Hence, GSTT2 expression has potential prognostic value for response to intravesical BCG treatment. Most importantly, we dissected the mechanisms of GSTT2 action in the host response to BCG treatment. We found that the GSTT2 gene influences intracellular BCG survival and reactive oxygen species (ROS), leading to variances in immune response after BCG exposure.

## 2. Results

### 2.1. GSTT2 Promoter Deletion and Its Impact on GSTT2 Expression and Cellular ROS

We investigated GSTT2 expression in human tissues by using the GTex database. This gene was expressed in most tissues, including the bladder, with the highest expression in the prostate, skin, thyroid, and adrenal glands (Appendix A). A panel of human bladder cancer and promonocytic cell lines was examined for GSTT2B deletion status [15] (Appendix A). GSTT2 expression post-BCG stimulation was analysed by real-time PCR. Cell lines with the GSTT2B^FL/Del^ genotype (THP-1) had a Ct value of 30.65, cell lines with the GSTT2B^FL/FL^ genotype (T24 and MGH) had Ct values of 25 and 24.2, and those with the GSTT2B^Del/Del^ genotype (UMUC3 and U937) had Ct values of 36.75 and 35, where Ct values above 32 were negative for gene expression (Appendix A). Thus, GSTT2 mRNA expression correlates with the GSTT2B status of genomic DNA.

Two cell lines, MGH (GSTT2B^FL/FL^) and UMUC-3 (GSTT2B^Del/Del^), were evaluated for the impact of GSTT2 expression on BCG-induced ROS [24]. GSTT2 expression in MGH cells was blocked using siRNA against GSTT2 (Figure 1A). A plasmid containing GSTT2 was transfected into UMUC-3 cells to increase GSTT2 expression (Figure 1B). Blocking GSTT2 expression decreased the percentage of cells with high ROS levels (Figure 1C) following exposure to lyophilised BCG. The opposite result was observed in UMUC-3 cells overexpressing GSTT2 **(**Figure 1D). Thus, GSTT2 modulates BCG-related ROS changes in cells.

### 2.2. Monitoring GSTT2 Impact on Human Bladder Cancer Cells and BCG

Lyophilised BCG was used for BCG toxicity and ROS generation assays, while live BCG was used for intracellular BCG survival experiments. GSTT2 knockdown or overexpression did not affect MGH or UMUC3 cell survival after exposure to BCG (Figure 1E,F), but it affected BCG survival within the cell lines (Figure 1G–J). Inhibition of GSTT2 expression in MGH cells increased BCG survival at 2 h post-exposure. However, after 24 h, this difference was not observed (Figure 1G). This observation was confirmed using a live/dead bacterial staining assay to evaluate BCG in MGH cells (Figure 1J). The expression of GSTT2 reduced BCG survival in the UMUC3 and U937 cell lines (Figure 1H,I). Using MGH cells, we observed increased NO production after a short exposure to BCG, which was reduced when GSTT2 was blocked, and this was linked to an increase in TNFα production (Appendix A). Given the impact of GSTT2 on cellular ROS, we monitored cellular DNA damage using a comet assay. There was reduced DNA damage in MGH cells with GSTT2 knockdown (Figure 1K,L), indicating that regular GSTT2 expression leads to DNA damage through its cellular function as blocking it reduces DNA damage. In UMUC3 cells, the impact of GSTT2 expression on DNA damage was unclear, as the empty vector itself induced DNA damage (Appendix A).

### 2.3. GSTT2B, GSTT2, and GSTT1 Analysis in Controls and Patients

To study the link between GST genotypes, risk of cancer development, and treatment response, GSTT2B deletion, GSTT2 single-nucleotide polymorphism, and GSTT1 deletion genotypes were analysed in a cohort of healthy individuals (*n* = 150) and bladder cancer patients (*n* = 205). Control and patient characteristics are presented in Table 1. There was no significant difference in the percentage of subjects with GSTT2B (GSTT2B^FL/FL^, GSTT2B^FL/Del^, and GSTT2B^Del/Del^) or GSTT1 (GSTT1^FL/FL^, GSTT1^FL/Del^, and GSTT1^Del/Del^) genotypes between the control and patient populations (Appendix A). Thus, loss of GSTT2 or GSTT1 expression was not associated with bladder cancer risk. GSTT2 promoter SNPs were not associated with bladder cancer development. The -627AA genotype was correlated with the GSTT2B^FL/FL^ genotype, and the -627GG genotype with the GSTT2^Del/Del^ genotype (*p* < 0.001) (Appendix A). The -537GG genotype was correlated with GSTT2B^FL/FL^ and the -537AA genotype with GSTT2B^Del/Del^ (*p* < 0.001). Expression of GSTT2B^FL/FL^ and GSTT1^Del/Del^ genotypes were significantly correlated (*p* < 0.001) (Appendix A). However, the incidence of GSTT2B^Del/Del^ subjects being GSTT1^FL/FL^ or GSTT1^Del/Del^ was similar.

### 2.4. GSTT2B, GSTT1, and GSTT2 SNPs Status and the Response to BCG Therapy

The incidence of recurrence, progression, and survival with reference to GSTT2B and GSTT1 status and GSTT2 SNPs was determined (Table 2 and Appendix A). Although fewer patients with GSTT2B^Del/Del^ and GSTT2B^FL/Del^ genotypes had recurrence than those with the GSTT2B^FL/FL^ genotype, the difference was insignificant. The basic therapy schedule for BCG therapy at NUH is ‘6+3’ instillations, with further repeated instillations for BCG failures. Not all the patients completed this schedule. Hence, some patients received fewer instillations than others. Based on the hypothesis that better BCG survival would lead to better immune activation, we divided the patients into two groups: those who received 9 or more (≥9) BCG instillations and those who received 8 or fewer (≤8) BCG instillations (approximately 74% had 6 instillations) (Appendix A) and evaluated their outcomes based on the GSTT2B and GSTT1 genotypes. We found that patients with the GSTT2B^Del/Del^ genotype were less likely to have a recurrence (0.041) with ≤8 BCG instillations, whereas the converse was true for those with the GSTT2B^FL/FL^ genotype (Appendix A). Comparing the incidence of recurrence in the GSTT2B^Del/Del^ cohort, those who received less therapy performed better (*p* = 0.028). GSTT2B genotypes did not modulate progression. The GSTT1 genotypes (Appendix A) and SNPs at -627, -537, and -277 did not correlate with recurrence or progression (Appendix A). Patients with the GSTT2B^Del/Del^ genotype also had better survival rates with less therapy (*p* < 0.02).

Based on Kaplan–Meier analysis, there was no significant difference in the overall recurrence-free survival between patients with different GSTT2B genotypes who received BCG therapy (Table 2 and Figure 2A). However, 89% of GSTT2B^Del/Del^ patients with recurrence had recurrence within 2 years of therapy commencement compared to those with the GSTT2B^FL/FL^ or GSTT2B^FL/Del^ genotypes (Figure 2A). Furthermore, Kaplan–Meier analysis based on the number of BCG instillations patients received indicated no difference in the outcomes between those who received ≤8 instillations of therapy and those who received ≥9 instillations of therapy (Figure 2B). However, when the patients were segregated by their GSTT2B genotypes, subjects who were GSTT2B^Del/Del^ had a significant difference in time to recurrence and survival between those who received ≤8 instillations and those who received ≥9 instillations (Figure 2C–E).

We analysed the TCGA database using GEPIA2 software (http://gepia2.cancer-pku.cn/#index, accessed on 7 April 2022) [25] to evaluate GSTT2 mRNA expression in bladder cancer and normal tissues (Figure 2F), and a wide variation in GSTT2 expression was found. Low GSTT2 expression correlated with better disease-free survival (Figure 2G). Based on our mRNA analysis of the cancer cell lines, we expected that low GSTT2 expression would correlate with the GSTT2B^FL/Del^ and GSTT2B^Del/Del^ genotypes. Low GSTT2 expression impacted disease-free survival for non-papillary tumours but not papillary tumours (Figure 2H,I). This observation was consistent with our multivariate analysis findings that GSTT2B^FL/FL^ and CIS were correlated with recurrence (Appendix A). Multivariate analysis of GSTT2B^FL/FL^ showed that it was associated with an increased risk of recurrence, with a hazard ratio of 5.5 (*p* = 0.002). Other predictive factors for recurrence were concomitant CIS (*p* = 0.003) and a history of multiple recurrences (*p* = 0.008). Regarding progression, only a history of recurrence was significant, and the GSTT2-537AA genotype was associated with an increased risk of progression (*p* = 0.036). Only age at diagnosis and smoking (univariate analysis) were significant for overall survival.

### 2.5. GSTT2 Modulates the DC Response to BCG

To gain insights into the role of GSTT2 in immune modulation in a mammalian system, we isolated DCs from the bone marrow of GSTT2KO and WT mice and stimulated them with BCG in vitro. Unstimulated DCs from GSTT2KO mice had significantly higher basal CD80 and CD86 expression and greater expression of MHC class II and CD86 proteins per cell, which appeared more activated than WT DC (Table 3). GSTT2KO DCs also produced significantly more IL12p70 and IL10 than WT DCs without stimulation (Table 3). Post-BCG stimulation (24 h), IL6 production from GSTT2KO mice was three-fold higher than that of WT DCs (Table 3). However, WT DCs produced more IL12p70 and TNFα than GSTT2KO DCs. DCs from both genotypes produced similar levels of IL10. Thus, there was a greater stimulation of inflammation after exposure to BCG in GSTT2KO DCs. GSTT2KO DCs were exposed to BCG for 2 h, treated as before to remove external BCG, or left in contact with BCG for 24 h. There was a similar increase in IL12p70, IL6, and IL10, but TNFα levels increased seven-fold after 24 h compared to 2 h of BCG exposure (1L12p70: 24.5 ± 6.2 pg/mL, IL6: 29,385.5 ± 3639.1 pg/mL, IL10: 3695.8 ± 534.3 pg/mL, and TNFα: 699.3 ± 103.3 pg/mL). The murine data indicates that GSTT2 absence results in more activated immune cells and a more robust inflammatory response to BCG. This may explain why patients with GSTT2B^Del/Del^ genotype responded better to fewer BCG instillations.

## 3. Discussion

In our current study, we find that GSTT2 expression has an influential impact on the intracellular survival of BCG in host cells. The theta class GST isoforms differ from other cytosolic GSTs in excluding 1-chloro-2,4-dinitrobenzene (CDNB) as a substrate, which is a general substrate for other GSTs [26]. However, secondary lipid peroxidation products and organic hydroperoxides were identified as GSTT2 substrates [27]. GSTT2 may exert its BCG elimination role by regulating the intracellular levels of secondary lipid peroxidation products. Our previous work demonstrated that internalising live BCG by bladder cancer cells leads to increased cellular ROS levels and lipid peroxidation, with lyophilised BCG inducing opposite effects [28]. GSTT2 expression was not upregulated after live BCG treatment. In contrast, GSTT2 significantly increased after lyophilised BCG treatment. It is plausible that lyophilised BCG treatment upregulates GSTT2 expression to detoxify lipid peroxidation products. In contrast, GSTT2 expression was not upregulated by live BCG treatment, leading to increased ROS and subsequent accumulation of lipid peroxidation products [28]. Patel et al. found that BCG has enhanced susceptibility to cumene hydroperoxide, an organic hydroperoxide, due to the loss of the organic hydroperoxide reductase (ohr) gene [29]. The ohr protein is a thiol-dependent peroxidase that acts on organic peroxides and peroxynitrites, two oxidants mainly involved in host–pathogen interactions. Hence, they are key players in bacteria defense against the host organic peroxide system [30]. Thus, GSTT2 expression has a profound effect on BCG survival in host cells, possibly via the modulation of secondary lipid peroxidation products. Intracellular BCG survival may affect the host immune response, leading to variances in BCG response in bladder cancer treatment, which will be discussed in another paragraph.

We also found that GSTT2 expression regulates BCG-induced ROS generation. Overexpressing GSTT2 in vitro increased net ROS after lyophilised BCG treatment and reduced intracellular live BCG survival (Figure 3). Hence, after live BCG treatment, GSTT2 overexpression may have exhausted the intracellular GSH supply for nullifying lipid peroxidation effects. Consequently, the depleted GSH results in a higher net ROS, aiding in the destruction of BCG particles via ROS. The opposite scenario is hypothesised for GSTT2-silenced cells, where intracellular GSH is not depleted by GSTT2 activities after live BCG treatment, leading to a decreased net ROS and increased BCG survival. In GSTT2-silenced cells, other enzymes, such as the alpha class GSTs, can perform the compensatory role of breaking down lipid peroxidation products. It is currently unclear if GSTT2-silenced cells can also resist delivering the internalised BCG in the phagosomes to lysosomes for terminal degradation. Furthermore, we did not detect lipid peroxidation products in this study. Hence, we have yet to determine if BCG elimination was facilitated by ROS or lipid peroxidation products. Nevertheless, BCG survival is transitory, as it was similarly reduced by 24 h in the presence or absence of GSTT2 (Figure 2G–I). Still, the initial increased BCG survival at 2 h may have led to a better inflammatory immune response in GSTT2KO cells.

This effect of GSTT2 on BCG survival was reminiscent of the effect of NRAMP1 (natural resistance and macrophage protein 1) on BCG survival. In mice, NRAMP1 modulates BCG survival and response to BCG immunotherapy [31], as shown by a natural single-amino-acid mutation that blocks NRAMP1 function [32]. In humans, evidence for the role of NRAMP1 in response to BCG immunotherapy has not been conclusive [33,34,35]. Given the frequent loss or reduction in GSTT2 expression in humans due to GSTT2B^Del/Del^ [15], a similar situation to that observed in mice with NRAMP1 exists in humans.

Our patients were usually treated with ‘6+3’ BCG instillations, but some patients could not tolerate this schedule and received fewer instillations of BCG. Treatment lapses often occur because patients have a robust inflammatory response to BCG therapy or develop cystitis. By analysing the response of this cohort of patients, we found that patients with the GSTT2B^Del/Del^ genotype who received ≤8 instillations of BCG were more likely to respond to therapy. Thus, the effect of GSTT2 on BCG survival in vitro correlated with the response to BCG immunotherapy. It is unknown why some GSTT2B^Del/Del^ patients responded to ≥9 instillations of BCG; however, this observation may indicate that polymorphisms in other genes might be involved in immune response modulation, such as cytokine gene polymorphisms.

In the literature, GSTT2 promoter SNP -537G has exhibited better transcriptional factor binding and protection against colon cancer development [19]. Thus, the G allele was expected to correlate with recurrence. However, no association was found. The combination of -537G, -277C, and -158T has also been associated with greater promoter activity [36]. In our population, subjects had a T at -158, and the majority had a T at -277, so these SNPs had no predictive value. Only GSTT2B status was correlated with outcomes. Our results differ from those of Kang et al., as we did not find an effect of GSTT1 on response to BCG therapy [21]. This difference could be because the GSTT1 null genotype occurs in 68% of their population but only in 39% of our patients. Our screening of the TCGA database revealed that GSTT2 expression in bladder cancer tissues is variable, consistent with our results. Analysis of overall patient survival based on GSTT2 expression levels showed better survival with lower GSTT2 expression.

GSTT2KO and WT mice were used to assess the ability of immune cells to respond to BCG in the absence and presence of GSTT2 without transient transfection, as well as the impact of GSTT2 on unstimulated cells. There was increased IL6 production, indicating enhanced inflammation, in response to BCG, which may explain why patients with GSTT2B^Del/Del^ could not tolerate repeated BCG dosing. High cytokine production might have led to a tolerogenic response [37] to repeated therapy, which would be responsible for poorer outcomes in the cohort with more treatment. Tolerogenic responses after repeated stimulation have been demonstrated in mice using LPS and peptidoglycan as stimulants [38]. GSTT2WT DCs produced more 1L12p70 and TNF-α after a 2 h BCG treatment than GSTT2KO DCs. These two cytokines are important in controlling human mycobacterial diseases [39,40]. In vitro, GSTT2 overexpression in bladder cancer cells decreased intracellular BCG survival. Thus, it is plausible that there is also a concomitant decrease in intracellular BCG survival in GSTT2 WT DCs, and the cytokine production in GSTT2 WT cells is geared towards rapid BCG clearance.

Rouanne et al. showed that BCG had direct effects on tumour tissues, which led to the downregulation of HLA-1 and EpCAM expression after BCG exposure, which was associated with disease progression in patients [41]. However, tumours that remained HLA-1^+^ showed an increased release of T-cell chemotactic factors [41]. After live BCG treatment, HLA-1 downregulation was demonstrated in UMUC3 and RT4 cell lines, which were GSTT2B^Del/Del^ and GSTT2B^Fl/Del^, respectively. Hence, GSTT2B status may not only impact immune function but could also affect tumour cell response to BCG treatment.

Our study shows that modulation of patient responses based on genotype may be achievable. Genotyping can also be used to reduce surveillance regimens for patients post-BCG therapy. For example, some patients in the GSTT2B^Del/Del^ group given ≤8 instillations (Figure 2C) had a recurrence within 2 years. Still, the remaining patients who did not experience a similar recurrence will remain disease-free for five years. Our data also indicated that subjects with GSTT2B^FL/FL^ genotypes responded poorly to BCG therapy and may be candidates for alternative treatments. This strategy could help ensure that BCG is administered to patients who can respond favourably. To put the importance of this finding in context, comparing populations, 25% of Singaporeans, 29% of Japanese, and 41.7% of Caucasians have the GSTT2B^Del/Del^ genotype [15].

This study has two main limitations. First, the patient cohort used for analysis is small and retrospective in nature. This study must be repeated with a larger cohort in a prospective longitudinal manner to substantiate the current findings. A larger population study could increase the confidence in using the GSTT2 genotype as a tool for patient stratification. Second, very little prior research exists regarding the effects of the GSTT2 genotype on bacteria–host interactions. Most studies report the association of the GSST2 gene polymorphisms with the susceptibility to malignant diseases. There is very little delineation of the mechanisms of GSTT2 action in the context of the risk of disease development or treatment response. Nevertheless, our work has revealed, to a certain extent, the areas in which GSTT2 contributes to BCG response in NMIBC immunotherapy, which are ROS regulation and intracellular bacteria survival.

## 4. Materials and Methods

### 4.1. Mammalian Cell Culture

All culture media were supplemented with 10% heat-inactivated fetal bovine serum, 2 mM L-glutamine, 50 U/mL penicillin G, and 50 µg/mL streptomycin (all reagents from Hyclone™, GE Life Sciences, Chicago, IL, USA) unless otherwise stated. Human transitional cell carcinomas MGH, T24, J82, RT4, THP1, and U937 (RRID: CVCL_0007). All cell lines were obtained from ATCC, Manassas, VA, USA, and were maintained in RPMI (Hyclone™, GE Life Sciences, Chicago, IL, USA). HEK293T cells (ATCC, Manassas, VA, USA) were maintained in DMEM (Hyclone™, GE Life Sciences, Chicago, IL, USA) supplemented with 2 mM sodium pyruvate. UMUC3 (ATCC, Manassas, VA, USA), UMUC6, UMUC9, and UMUC14 (kind gifts from Dr. HB Grossman of MD Anderson Cancer Center, TX, USA)) were cultured in MEM (Gibco, Thermo Fisher, Waltham, MA, USA). HUC-1 cells (RRID: CVCL_3798, ATCC) were maintained in Ham’s F-12 Nutrient Mixture (Gibco, Thermo Fisher, Waltham, MA, USA). All cell lines were maintained at 37 °C in a 5% CO_2_ atmosphere and routinely passaged when 85–90% confluent. Cell lines (except for UMUC9 and UMUC14) were authenticated using short tandem repeat genotyping with the Cell ID™ system (Promega, Madison, WI, USA) according to the manufacturer’s protocol. All cell lines tested were between 80 and 100% matched with the ATCC cell line database.

### 4.2. BCG Preparation and Maintenance

Lyophilised and live BCG, Connaught strain, were prepared as previously described [28]. Lyophilised BCG (ImmuCyst^®^, Aventis Pasteur, ON, Canada) was used to prepare live BCG by plating on Middlebrook 7H10 (BD, Franklin Lakes, NJ, USA) supplemented with 10% ADC (0.85% NaCl, 5% bovine serum albumin fraction V, 2% dextrose, and 0.003% catalase), 0.05% Tween 80, and 0.2% glycerol (Sigma, St. Louis, MO, USA). Single colonies were selected, expanded, and maintained in Middlebrook 7H9 (BD, Franklin Lakes, NJ, USA) broth. Live bacteria were harvested from the exponential phase (OD_600nm_ of 0.7–0.8), and the formula OD_600nm_ 0.1 = 2.6 × 10^6^ colony-forming units (cfu)/mL was used to calculate the cfu of bacteria. Lyophilised BCG (81 mg) was prepared by resuspension in the provided buffer, followed by washing twice in 10× volume of PBS (137 mM NaCl, 2.7 mM KCl, and 10 mM phosphate pH 7.4) and centrifugation at 99× *g* for 3 min. The supernatant containing BCG was isolated. The OD440nm was measured, and the formula OD_440nm_ 0.2 = 3 × 10^6^ cfu/mL was used to determine the cfu of lyophilised BCG.

### 4.3. GSTT2 Gene Cloning

The plasmid bearing the human GSTT2 gene (pQEGSTT2) was a kind gift from Prof. PG Board (John Curtin School of Medical Research, Canberra, Australia). GSTT2 was excised by BamHI (ThermoScientific, Waltham, MA, USA) digestion and cloned into the same restriction site in pBUDCe4.1, a mammalian expression vector (Invitrogen), to generate pBUDGSTT2.

### 4.4. Transfection Protocol

The Dharmacon (Lafayette, CO, USA) ON-TARGETplus SMARTpool siRNA (L-011181-00-0005) system with a mixture of four duplex siRNAs targeting GSTT2 knockdown was used for GSTT2 silencing. Target sequences were J-11181-05(GCUCAAGGAUGGUGAUUUC), J-011181-06(GCACCGUGGAUUUGGUCAA), J-011181-07(AGGCUAUGCUGCUUCGAAU), and J-011181-08(ACACUGGCUGAUCUCAUG). ON-TARGET*plus* non-targeting siRNA #1 (NNUGGUUUACAUGUCGACUAA) was used as the non-targeting siRNA control. Transfection of siRNAs and plasmid DNA was accomplished using DOTAP (Roche, Basel, Switzerland). MGH and UMUC3 cells (2 × 10^5^) were plated overnight in 6-well plates before transfection. The cells were transfected with 2.5 µg of plasmid DNA or 30 pmol of siRNA complexed with 20 µg of DOTAP and 40 µg of methyl-β-cyclodextrin (MBC; Sigma, St. Louis, MO, USA) for 2 h. U937 cells (2 × 10^5^) were plated in 24-well plates and treated with 100 ng/mL phorbol 12-myristate 13-acetate (Sigma, St. Louis, MO, USA) for 48 h to induce cell differentiation. The cells were allowed to recover for 24 h before transfection. Cells were transfected with 1.25 µg DNA complexed with 4 µg DOTAP and 8 µg MBC for 2 h. Twenty-four hours later, the cells were exposed to BCG at a 1:100 (cell/bacteria) ratio for 2 h. The plates with BCG were centrifuged at 453× *g* for 5 min to maximise cell contact with the bacteria.

### 4.5. PCR Validation of GSTT2 Expression

RNA was extracted from transfected cells after 24 h using TRIzol^®^ reagent (Invitrogen, Carlsbad, CA, USA), and approximately 5 µg of RNA was converted to cDNA using SuperScript™ II Reverse Transcriptase (Invitrogen, Carlsbad, CA, USA). The cDNAs were used for the PCR validation of GSTT2 expression. PCR was performed with 5 µL cDNAs as template, 0.5 µM of GSTT2 (forward: 5′AGGCTCGTGCCCGTGTTC 3′, reverse: 5′ GGCCTCTGGTGAGGGTG 3′) or RPS27A (forward: 5′CTCGAGGTTGAACCCTCG 3′, reverse: 5′ GCACTCTCGACGAAGGCG 3′) specific primers, 0.2 mM of dNTPs, 0.5 U of DyNAzyme™ DNA polymerase (ThermoFisher, Waltham, MA, USA) in 1x buffer (10 mM Tris-HCl pH 8.8, 50 mM KCl, 1.5 mM MgCl_2_, 0.1% Triton^®^ X-100) as described before [11]. The amplified products were separated on a 1.5% agarose gel. For real-time PCR, predesigned TaqMan® Gene Expression Assay probes (Applied Biosystems, Waltham, MA, USA) for GSTT2 (Hs00168315-m1) and endogenous control GAPDH (Hs99999905-m1) were used with a 1 FAM™ dye-labelled TaqMan® MGB probe and analysed on a 7500 Real-Time PCR System (Applied Biosystems, Waltham, MA, USA).

### 4.6. Measurement of Cellular ROS and Comet Assay

Cells were washed three times with ice-cold 1x PBS and scraped in a final 1 mL 1xPBS. The cells were filtered through a 60 µ nylon microfilament. For each sample, 10 µL of a 1 mg/mL concentration of H2DCF-DA (ThermoFisher, Waltham, MA, USA) in DMSO (Sigma, St. Louis, MO, USA).) was added, and tubes were vortexed gently to mix. The cells were incubated at 37 °C for 10 min before immediate flow analysis. The fluorescence signal was detected using the FITC channel at an excitation wavelength of 488 nm and emission wavelength of 535 nm using a FACS Canto™ flow cytometer (BD, Franklin Lakes, NJ, USA). The Comet Assay was performed as described by Bajpayee et al. [42]. Three microscope fields were examined at 4× magnification, and a minimum of 20 comets were analysed per sample. The tail moments were analysed using CometScore Freeware v 1.5 (Tritek Corporation, Sumerduck, VA, USA).

### 4.7. Cytotoxicity and BCG Survival Assay

After live BCG treatment, the cells were washed twice with 1x PBS and treated with 200 µg/mL gentamicin (Gibco, Thermo Fisher, Waltham, MA, USA) for 2 h to kill any extracellular BCG. After washing (twice with 1×PBS), the cells were incubated in fresh media for a further 48 h (for cytotoxicity assay), immediately lysed, or left for 24 h (for BCG survival assay). Cytotoxicity was determined by counting viable cells using a hemocytometer. The assay was performed by lysing the cells in 1 mL of Triton X-100 (Sigma, St. Louis, MO, USA, 0.1% *v*/*v* in water) for 5 min at 37 °C. The lysates were serially diluted in 1×PBS with 0.05% Tween80 (Sigma, St. Louis, MO, USA) to a 10^−2^ dilution, and the lysates (0.1 mL) were spread on Middlebrook 7H10 agar plates in triplicate. The plates were incubated at 37 °C with 5% CO_2_ until colonies were visible for counting (3–4 weeks). An intracellular bacterial survival assay using the Live/Dead™ BacLight™ kit (Invitrogen, Carlsbad, CA, USA) was performed by lysing the cells in 1 mL of 0.1% deoxycholate for 5 min, followed by the addition of 3 µL of dye mixture (1:1 ratio of components A and B) for 15 min in the dark. The fluorescence signals were detected, and images were captured with an Olympus FluoView FV1000 (Olympus, Tokyo, Japan) laser scanning confocal microscope using a 60× oil (1.45) objective with 488 nm argon ions and a HeNe (543 nm and 633 nm) laser as the excitation and emission sources. Fluorescence levels were analysed by measuring the integrated density using the ImageJ v1.53 (RRID: SCR_003070) analysis software.

### 4.8. Patient Samples

Informed consent from the patients and healthy subjects was obtained, as well as approval by the Domain-Specific Review Board (NHG DSRB:2012/00475, NHG DSRB:2018/00197, and CIRB 2011/061/B). Genomic DNA was isolated from the blood samples using a DNeasy Blood and Tissue Kit (Qiagen, Hilden, Germany). Blood was collected from high-risk NMIBC patients (both male and female) who had either been enrolled in trials evaluating BCG therapy (*n* = 111) or were non-trial patients receiving BCG therapy (*n* = 94). The patients received either standard-dose BCG (81 mg, Immucyst^®^, Aventis Pasteur, ON, Canada), low-dose BCG (27 mg), or low-dose BCG plus IFN-α treatment in a “6+3” schedule comprising six once-weekly instillations and three once-weekly instillations with a 6-week break between the two courses. Some patients received further BCG instillations consisting of 3 instillations repeated at 3-month intervals based on physician discretion or patient acceptability for up to 3 years. Some patients received less than nine instillations for reasons of their health condition, toxicity associated with therapy, or the decision of the clinical group. The BCG strains used in clinical treatment were BCG Connaught, Tice, and Tokyo strains. However, most patients were treated with BCG Connaught. Healthy, age-matched subjects or volunteers were recruited during routine health checks at the National University Hospital (NUH).

### 4.9. GSTT2 Promoter Deletion Analysis

The triple primer PCR method described by Zhao et al. [15] was used to detect GSTT2B deletions. GSTT1 deletions were detected by two PCR assays. The PCR reaction was performed in a final volume of 10 µL with 0.5 U Taq polymerase (GoTaq, Promega, Madison, WI, USA), 1.5 mM MgCl_2_, 2 mM dNTPs, 10 pmol of each primer, and 10 ng of genomic DNA. The thermal cycling conditions were as follows: denaturation at 95 °C for 2 min; 30 cycles of denaturation at 95 °C for 30 s, annealing at 60 °C for 30 s, and extension at 72 °C for 45 s; and a final extension step at 72 °C for 5 min. PCR products were separated on 1% agarose gel. A known 17bp duplication in the GSTT2 promoter was analysed by polyacrylamide gel electrophoresis of the PCR products. SNPs at -627, -537, and -277 were analysed by performing PCR with Type-It High-Resolution Melt (HRM) master mix (containing EvaGreen, a fluorescent dye that binds double-stranded DNA) on 10 ng DNA and with 0.5 μM primers for 50 cycles, followed by HRM analysis using the Rotor-Gene Q 2 plex real-time PCR machine (Qiagen, Hilden, Germany). HRM melt curve profiles were generated, and melt curve aberrations, indicative of a single sequence variant, were confirmed by sequencing. Sequence information was input into the HRM analysis software v2.0.2 (Build 4) to assign SNP sequences based on the melt profile. Appendix A lists the primer sequences and annealing temperatures used in this study.

### 4.10. GSTT2 Knockout (KO) Mice

Animal studies and breeding protocols were approved by the Institutional Animal Care and Use Committee of the National University of Singapore (protocols BR17-1422 and R17-1434). GSTT2KO mice were generated using the CRISPR system by deleting exon 2 of the murine GSTT2 gene (Transgenic Mice Facility, Cancer Science Institute (CSI)). Mice used for the study were from the 3rd to 4th generation, produced by mating heterozygous mice. Both male and female mice were used in this study. Bone marrow-derived DCs from GSTT2 KO and wild-type (WT) mice were generated from the femurs of mice as described previously [43] and were exposed to live BCG for 2 h and 24 h. Cytokine production (IL12p70, IL6, IL10, and TNFα) was examined by ELISA (eBioscience™, Thermo Fisher Scientific, Waltham, MA, USA). Immune cell activation was determined based on the expression of MHC class II, CD80, CD86, CD40, and CD83, as measured by flow cytometry. APC anti-mouse CD11c, FITC anti-mouse CD80, FITC anti-mouse I-Ab, and FITC anti-mouse H-2Kb/H-2Db were obtained from BioLegend (San Diego, CA, USA). PE anti-mouse CD83, PE anti-mouse CD86, PE anti-mouse CD40, PerCP-Cy5.5 anti-mouse Ly6G, and FITC anti-mouse CD11b antibodies were purchased from eBioscience.

### 4.11. Statistical Analysis

For in vitro studies, one-way ANOVA with the post hoc Bonferroni test was used to compare three or more samples. Independent sample t-tests were used to compare two samples. A *p*-value <0.05 was considered to indicate a statistically significant difference. All in vitro studies were performed twice in triplicates (*n* = 6). The time to first recurrence, progression, and death were analysed using the Kaplan–Meier method. COX regression analysis was performed to assess the impact of genotype and treatment. A frequency distribution histogram was plotted to observe the distribution of the data. All analyses were performed using the IBM SPSS statistical analysis software version 24.0 (RRID: SCR_002865).

## 5. Conclusions

Here, we have identified a gene, GSTT2, responsible for intracellular BCG survival and ROS regulation in the BCG immunotherapy of bladder cancer. GSTT2 genotyping is a potential tool for identifying patients who will not respond well to BCG immunotherapy and can respond favourably to fewer BCG instillations. This study underscores the need to find suitable genetic markers to be used as indicators for BCG treatment response. Although it was not the focus of the current work, future endeavours to study the polymorphisms of the various genes involved in detoxification and redox response via a genome-wide association study approach should be considered. Findings from such investigations can expedite advancements in the impact of evidence-based genomics for clinical oncology management.

## Figures and Tables

**Figure 1 ijms-25-08947-f001:**
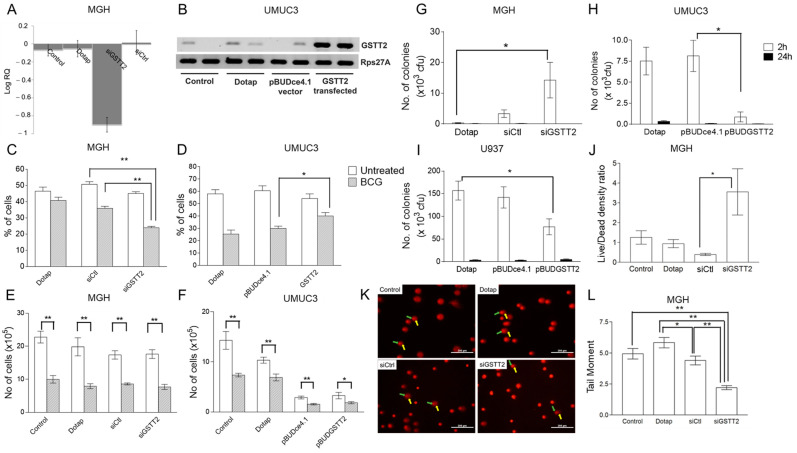
GSTT2 expression in bladder cancer cell lines and its impact on cellular ROS, cytotoxicity, BCG survival, and DNA damage after exposure to BCG. (**A**) Real-time analysis of GSTT2 gene expression after silencing with siGSTT2 and siControl in MGH cells. (**B**) RT-PCR detection of GSTT2 gene expression in UMUC3 cells after transfection with a plasmid carrying the GSTT2 gene, pBudGSTT2, and pBudCE4.1 (empty vector). (**C**) The impact of GSTT2 downregulation on the percentage of MGH cells with high ROS (measured with H_2_DCFDA) after exposure to lyophilised BCG. (**D**) The impact of GSTT2 expression on the percentage of UMUC-3 cells with high ROS levels after stimulation with lyophilised BCG. A *p*-value < 0.05 is designated as * and < 0.001 as **. (**E**,**F**) The cytotoxic effect of BCG on MGH and UMUC3 cells in the presence and absence of GSTT2 expression. ** represents a value < 0.001 and * *p*-value < 0.05. (**G**–**J**) Intracellular survival of live BCG in (**G**) MGH, (**H**) UMUC3, and (**I**) U937 cells after blocking or overexpression of GSTT2. The cells were GSTT2 silenced or induced to overexpress GSTT2, followed by a 2 h exposure to BCG, removing excess BCG for the 24 h analysis. The number of BCG colony-forming units that survived was determined by lysing the cells and plating the BCG on agar plates and (**J**) performing a live/dead bacteria assay on MGH cells. * represents a *p*-value < 0.05. (**K**) Photos of DNA comets observed in MGH cells with and without GSTT2 silencing. The green arrows indicate the nucleus head and the yellow arrows indicate the comet trail comprising damaged DNA. (**L**) Graph showing the calculated tail moments using CometScore v1.5. Experiments were performed twice in triplicate (*n* = 6). A minimum of 20 comets were analysed per sample. * represents a *p*-value < 0.05 and ** a *p*-value < 0.01.

**Figure 2 ijms-25-08947-f002:**
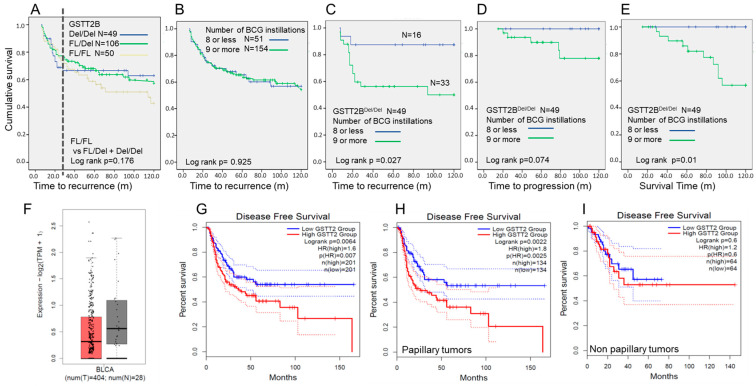
Clinical impact of GSTT2B and GSTT2 in bladder cancer. (**A**–**E**) Kaplan–Meier plots showing cumulative survival with time. (**A**) Time to recurrence segregated by GSTT2B genotypes. (**B**) Time to recurrence segregated by the number of BCG instillations received (i.e., ≤8 versus ≥9), (**C**) time to recurrence, (**D**) progression, and (**E**) survival for subjects with the GSTT2B^Del/Del^ segregated by the number of BCG instillations received. The Log-rank *p*-value is noted on each graph, and the number of samples (N) is indicated. The dotted line on graph (**A**) indicates that most recurrences in the cohort of patients with the GSTT2B^Del/Del^ genotype occurred within 2 years of starting therapy. (**F**) Data extracted from the TCGA database for GSTT2 expression levels in tumour and normal bladder tissues, (**G**) disease-free survival with respect to high and low GSTT2 mRNA expression in tumour tissue, and (**H**) disease-free survival in papillary and (**I**) non-papillary tumours with respect to GSTT2 expression levels.

**Figure 3 ijms-25-08947-f003:**
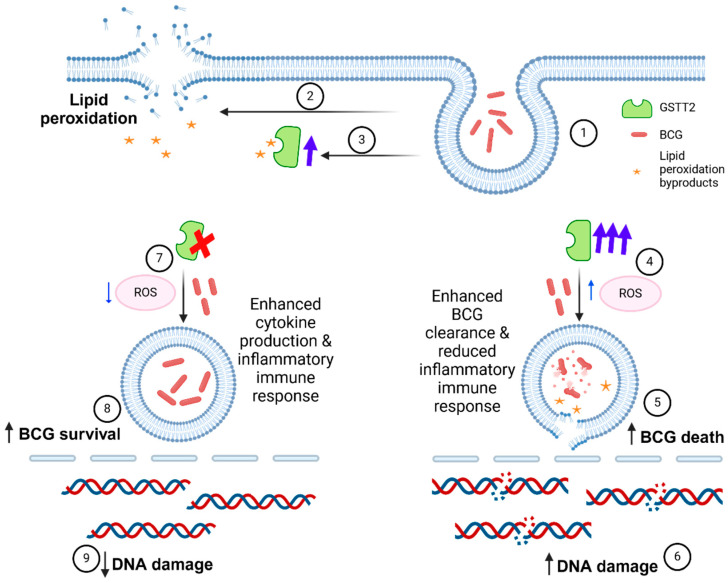
Schematic of the impact of GSTT2 expression on BCG immunotherapy of bladder cancer. (1) The internalisation of BCG particles (2) induces lipid peroxidation and (3) GSTT2 expression upregulation to catalyse the breakdown of lipid peroxidation by-products. (4) When GSTT2 was transiently overexpressed, lyophilised BCG treatment, which usually induced ROS reduction, resulted in a higher net ROS level than normal cells without GSTT2 expression treated similarly with BCG. (5) BCG survival was impaired in GSTT2 overexpressed bladder cancer cells, possibly due to the generation of lipid peroxides (from bacteria or host cells) and increased ROS. The enhanced BCG clearance in these cells reduces subsequent inflammatory immune response. (6) GSTT2 overexpression led to a baseline increase in DNA damage without BCG treatment. (7) Lyophilised BCG treatment of GSTT2-silenced cells led to a lower net ROS level than normal GSTT2-expressing cells treated similarly with BCG. (8) There was significantly more BCG surviving intracellularly in GSTT2-silenced cells, which enhanced inflammatory response and cytokine production. (9) GSTT2 silencing led to a baseline decrease in DNA damage without BCG treatment. This image was created with BioRender.com.

**Table 1 ijms-25-08947-t001:** Control and patient characteristics.

General Characteristics	Controls (*n* = 149)	Patients (*n* = 205)	*p*-Value
Chinese/Malay/Indian/Others * %	75.8/10.1/9.4/4.7	79.4/7.8/6.4/6.4	NS
Males/Females %	76.5/23.5	82.0/18.0	NS
Age (years)	57.0 ± 10.26	63.80 ± 11.0	<0.001 **
Number of patients who received BCG instillations: fewer than 8/6 + 3 and more		51/154	
Number of BCG instillations: range/median number of instillations		4–27/9	
Patients with at least 10 years of follow-up data		69 (33.7%)	
Patients with at least 5 years of follow-up data		141 (68.8%)	
Patients with at least 3 years of follow-up data		177 (70.8%)	
Tumour stage: Ta/T1/CIS/CIS + Ta/CIS + T1/T2		46/75/22/12/25/1	
Tumour grade: G1/G2/G3		23/41/110	
Tumour grade: Low grade/High grade		6/25	
Therapy BCG: 81 mg/27 mg/27 mg + IFN α/unknown BCG dose		92/20/66/27	
Recurrence		80/205 (40.0%)	
Progression		24/205 (11.7%)	
Death: Cancer/Unrelated/Unknown		19 (10.7%)/39 (22.0%)/10 (5.6%)	
Smoking: Smokers/Non-smokers/Unknown		87/83/7	
Follow-up (years): Minimum − Maximum/Median		1.06–18.2/87.6	

* Caucasian or undeclared. NS—not significant. BCG—Mycobacterium bovis Bacillus Calmette–Guerin, CIS—Carcinoma in situ. ** Analysed by univariate statistics.

**Table 2 ijms-25-08947-t002:** GSTT2B and GSTT1 genotypes and response to therapy.

Genes		GSTT2B					GSTT1				
Response to Therapy/ Genotype		FL/FL	FL/Del	Del/Del	Total	*p*-Value	FL/FL	FL/Del	Del/Del	Total	*p*-Value
Incidence of recurrence											
No		25 (50.0%)	67 (63.2%)	31 (63.3%)	123(60.0%)	NS	15 (57.7%)	65 (67.7%)	42(52.5%)	122(60.4%)	0.116
Recurrence		25 (50.0%)	39 (36.8%)	18 (36.7%)	82 (40.0%)		11 (42.3%)	31 (32.3%)	38(47.5%)	80 (39.6%)	
≥9	No	19 (51.4%)	56 (66.7%)	18 (54.5%)	93	NS	13 (65.0%)	49 (66.2%)	31 (53.4%)	93	NS
	Recurrence	18 (48.6%)	28 (33.3%)	15 (45.5%)	61		7 (35.0%)	25 (33.8%)	27 (46.6%)	59	
≤8	No	6 (46.2%)	11 (50.0%)	14 (87.5%)	30	0.041	2 (33.3%)	16 (72.7%)	12 (54.5%)	30	NS
	Recurrence	7 (53.8%)	11 (50.0%)	2 (12.5%)	20		4 (66.7%)	6 (27.3%)	10 (45.5%)	20	
*p*-value by instillations		NS	NS	0.028			NS	NS	NS		
Incidence of Progression											
No		41 (82.0%)	96 (90.6%)	44 (89.8%)	181(88.3%)	NS	23(88.5%)	8(88.5%)	70 (87.5%)	178(88.1%)	0.976
Progression		9 (18.0%)	10 (9.4%)	5 (10.2%)	24 (11.7%)		3 (11.5%)	11(11.5%)	10 (12.5%)	24 (11.9%)	
≥9	No	31 (83.8%)	77 (91.7%)	28(84.8%)	136(88.3%)	NS	18(90.0%)	66 (89.2%)	50 (86.2%)	134(88.2%)	NS
	Progression	6 (16.2%)	7 (8.3%)	5 (15.2%)	18 (11.7%)		2 (10.0%)	8 (10.8%)	8 (13.8%)	18 (11.8%)	
≤8	No	10 (76.9%)	19 (86.4%)	16 (100%)	45 (88.2%)	NS	5 (83.3%)	19 (86.4%)	20 (90.9%)	44 (88.0%)	NS
	Progression	3 (23.1%)	3 (13.6%)	0 (0%)	6 (11.8%)		1 (16.7%)	3 (13.6%)	2 (9.1%)	6(12.0%)	
*p*-value by instillations		NS	NS	NS			NS	NS	NS		
Overall Survival											
Yes		39 (78.0%)	84 (79.2%)	39 (79.6%)	162(79.0%)	NS	19 (73.1%)	73 (76.0%)	67 (83.8%)	159(78.7%)	0.348
No		11 (22.0%)	22 (20.8%)	10 (20.4%)	43(21.0%)		7 (26.9%)	23 (24.0%)	13 (16.3%)	43 (21.3%)	
≥9	Yes	31(83.8%)	65(77.4%)	23(69.7%)	119(77.3%)	NS	13 (65.0%)	56 (75.7%)	48 (82.8%)	117(77.0%)	NS
	No	6 (16.2%)	19(22.6%)	10 (30.3%)	35 (22.7%)		7 (35.0%)	18 (24.3%)	10 (17.2%)	35 (23.0%)	
≤8	Yes	8 (61.5%)	19 (86.4%)	16 (100%)	43 (84.3%)	NS	6 (100.0%)	17 (77.3%)	19 (86.4%)	42 (84.0%)	NS
	No	5 (38.5%)	3 (13.6%)	0 (0%)	8 (15.7%)		0	5 (22.7%)	3 (13.6%)	8 (16.0%)	
*p*-value by instillations		NS	NS	0.02			NS	NS	NS		

NS—not significant. FL—Full Length. Del—Deleted. GSTT2B—Glutathione-S-transferase theta 2. Data were analysed by the Kaplan–Meier method.

**Table 3 ijms-25-08947-t003:** Immune activation and cytokine production of GSTT2KO and WT DC in the presence and absence of BCG.

	CD11c	Percentage of Cells		Mean Fluorescence Index (MFI)		Cytokines (pg/mL)		
Unstimulated		WT	KO	WT	KO		WT	KO
	CD40	9.9 ± 6.4	20.4 ± 20.8	72.8 ± 44.9	90.0 ± 100.5	IL12p70	4.0 ± 0.6	26.9 ± 4.5 *
	CD80	8.0 ± 4.7	19.5 ± 6.3 *	115.8 ± 101	155.3 ± 32.1	IL6	56.1 ± 13.6	53.1 ± 8.2
	CD83	14.6 ± 14.4	18.1 ± 15.1	158.5 ± 92.7	520.4 ± 111.3 *	IL10	70.1 ± 14.4	124.5 ± 45.9 *
	CD86	11.9 ± 6.4	35.2 ± 11.1 *	258.8 ± 225.3	1565.8 ± 1256.2 *	TNFα	168.0 ± 69.2	77.7 ± 10.2
	MHC II	32.5 ± 6.5	22.9 ± 12.0	357.8 ± 183.7	961.8 ± 263.4 *			
Lyophilised BCG	CD40	25.6 ± 5.2	22.7 ± 11.3	171.5 ± 51.6	211.8 ± 103.5	IL12p70	144.1 ± 2.2 ^†^	32.2 ± 5.4 *
	CD80	9.0 ± 2.5	17.0 ± 2.8	67 ± 21.2	270.5 ± 66.1 *	IL6	10537.3 ± 592.7 ^†^	31307.3 ± 4709.4 *^†^
	CD83	11.2 ± 8.8	17.2 ± 11.4	98 ± 39.6	305.9 ± 114.5	IL10	4909.6 ± 1207.6 ^†^	4513.4 ± 233.8 ^†^
	CD86	32.2 ± 0.4	43.4 ± 9.2	281 ± 0	1006.5 ± 311.0	TNFα	10569.1 ± 1339.2 ^†^	4095.8 ± 2776.5 *^†^
	MHC II	32.5 ± 0.7	38.6 ± 9.2 *	259.5 ± 12	1083.3 ± 126.5			

^†^ *p* < 0.05, compared to the unstimulated control; * *p* < 0.05, compared to the WT. Comparisons of means was performed using one-way ANOVA with Bonferroni post hoc test. WT—wild type, KO—knockout, CD—cluster of differentiation, MHC—major histocompatibility complex, IL—interleukin, TNF—tumour necrosis factor. Cytokine production was measured

## Data Availability

All research data supporting this publication are included in the manuscript and Appendix A.

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
