# Peer review of "Glutathione-S-Transferase Theta 2 (GSTT2) Modulates the Response to Bacillus Calmette–Guérin Immunotherapy in Bladder Cancer Patients"

_ijms, 2024, doi:10.3390/ijms25168947_

Round 1

Reviewer 1 Report

Comments and Suggestions for Authors

I read with interest the manuscript entitled "Glutathione-S-Transferase Theta 2 (GSTT2) modulates the response to BCG immunotherapy in bladder cancer patients"

The abstract is clear and well-conceived with all relevant information for the reader. Please add a few more keywords from MeSH terms to potentially expand the visibility of the manuscript.

At the end of the introduction, more clearly define the goal and hypotheses of your research. In relation to previous research on the mentioned topic, clearly point out why this study is new and worthy of attention.

The materials and methods section is well written so that the study can be reproducible.

In parentheses, you must indicate the country next to all manufacturers. For some you mentioned, but for some you didn't.

Within section 2.11. indicate which test you used to test the normality of the distribution of the obtained data.

Within all tables, using a superscript, next to the p-value, state which test you used. Also, under the tables, write the full name of all used abbreviations within the tables.

I suggest that table 3, table 4, and figure 3 be included in the suppl. materials.

I suggest that you start the discussion by presenting the most relevant results from your study, and include the parts that are more appropriate for an introduction in the introduction of the manuscript.

Please reconsider all the strengths and limitations of your study, and improve the strengths and limitations paragraph within the discussion.

Please follow the MDPI template and change the order of the chapters to place the materials and methods section at the end.

Accordingly, match the order of the references. Also, the references are not written in accordance with the instructions for authors. Please correct them.

Comments on the Quality of English Language

Moderate editing of English language required.

Author Response

The abstract is clear and well-conceived with all relevant information for the reader. Please add a few more keywords from MeSH terms to potentially expand the visibility of the manuscript.

Response: We have increased the number of MESH terms under the keywords.

At the end of the introduction, more clearly define the goal and hypotheses of your research. In relation to previous research on the mentioned topic, clearly point out why this study is new and worthy of attention.

Response: In the two final paragraphs of the introduction, we specifically outline our hypothesis and goal and emphasize the significance of our findings.

The materials and methods section is well written so that the study can be reproducible. In parentheses, you must indicate the country next to all manufacturers. For some you mentioned, but for some you didn't.

Response: The manufacturers’ details are all provided and standardized.

Within section 2.11. indicate which test you used to test the normality of the distribution of the obtained data.

Response: The frequency distribution test is added to the statistics section.

Within all tables, using a superscript, next to the p-value, state which test you used. Also, under the tables, write the full name of all used abbreviations within the tables.

Response: The statistical test used and the full form of each abbreviation is provided below each table

I suggest that table 3, table 4, and figure 3 be included in the suppl. materials.

Response: We have moved Table 3 to the Supplementary Information document as Supplementary Table 5. However, Table 4 (now Table 3) and Figure 3 will remain in the main text as these items contain important information that helps to improve the reading experience for the IJMS audiences.

I suggest that you start the discussion by presenting the most relevant results from your study, and include the parts that are more appropriate for an introduction in the introduction of the manuscript.

Response: We have removed some parts of the discussion and kept only relevant parts to improve the readability and make the section more concise.

Please reconsider all the strengths and limitations of your study, and improve the strengths and limitations paragraph within the discussion.

Response: We have expanded on this part of the discussion, which now encompasses a whole paragraph.

Please follow the MDPI template and change the order of the chapters to place the materials and methods section at the end.

Response: We have amended the document to follow the order of the MDPI template.

Accordingly, match the order of the references. Also, the references are not written in accordance with the instructions for authors. Please correct them.

Response: The references are formatted according to the MDPI endnote style template.

Reviewer 2 Report

Comments and Suggestions for Authors

According to the authors "This study evaluated the association of GSTT1, GSTT2, and GSTT2B with BCG response in non-muscle invasive bladder cancer treatment". I leave the results of the conducted research to the experts. I evaluate the work only from a statistical point of view. In my opinion, the statistical methods used for research are appropriate. In order to show the influence of GSTT2B and GSTT2 in bladder cancer, the authors even provide survival function graphs obtained by the Kaplan Meier method. Of course, I would prefer to see analytical expressions of survival functions that correspond to empirical data. In this case, the reader could calculate the characteristics of the remaining life span himself. It is quite difficult to do that from the graphs. I have some notes for work.

(1) The work abstract uses a whole series of abbreviations, for example: GSTT1, GSTT2, GSTT2B. It would be better not to use abbreviations in the  abstract. Even if the shortcut is described in the wiki.

(2) Long formulas (see lines 133-135 for instance) should be displayed on separate lines.

(3) Some places in the text (see lines 215, 219 for instance) are unnecessarily underlined. View full text.

(4) The internal labels of the graphs in Figures 1 and 2 are difficult to read. It should be highlighted.

(5) Use of bold should be revised throughout the text. Especially in tables.

(6) Reference list should be prepared in MDPI style.

Author Response

Reviewer 2

The work abstract uses a whole series of abbreviations, for example: GSTT1, GSTT2, GSTT2B. It would be better not to use abbreviations in the  abstract. Even if the shortcut is described in the wiki.

Response: We have amended the abstract, and the full form of the abbreviations was added.

Long formulas (see lines 133-135 for instance) should be displayed on separate lines.

Response: Long formulas are now displayed in separate lines.

Some places in the text (see lines 215, 219 for instance) are unnecessarily underlined. View full text.

Response: The underline is removed.

The internal labels of the graphs in Figures 1 and 2 are difficult to read. It should be highlighted.

Response: We have amended Figures 1 and 2 to improve the readability of the internal labels of each graph.

Use of bold should be revised throughout the text. Especially in tables.

Response: Each graph was checked, and the unnecessary bolded items were removed.

Reference list should be prepared in MDPI style.

Response: References are now formatted using the MDPI Endnote style template.

Reviewer 3 Report

Comments and Suggestions for Authors

This manuscript discusses the Glutathione-S-Transferase Theta 2 (GSTT2) modulates the response to BCG immunotherapy in bladder cancer patients. The present manuscript is good. However, some minor errors still need to be checked before possible publication in the “International Journal of Molecular Sciences”.

Comments to Authors:

-        Improve the abstract section.

-        Page 01, line 16: Need to abbreviate the “BCG” before using the abbreviation.

-        The introduction section should be revamped with the significance and importance of the presented work.

-        Add the scale bar to the microscopic images of DNA comets observed in MGH cells with and without GSTT2 silencing (Figure 1L).

-        Need to label the observed morphological changes in the Figure 1L.

-        Improve the quality of figure 2 plots.

-        Which factor more dominant in the impact of GSTT2 expression on BCG immunotherapy of bladder cancer? Explain it.

-        Authors need to cite the recent year references in the discussion part.

-        Conclusion section is missing in the present manuscript.

-        To improve the manuscript, the authors could compare their research with other studies in a table to highlight the benefits of their work.

-         Finally, the text should be checked for grammatical, formatting, and punctuation mistakes.

Author Response

Reviewer 3

Improve the abstract section.

Response: The abstract section was amended for improvement.

Page 01, line 16: Need to abbreviate the “BCG” before using the abbreviation.

Response: The abbreviation full form is included in the abstract.

The introduction section should be revamped with the significance and importance of the presented work.

Response: The final paragraph of the introduction emphasizes the significance and importance of the current study.

Add the scale bar to the microscopic images of DNA comets observed in MGH cells with and without GSTT2 silencing (Figure 1L).

Response: The scale bar was added to all the comet assay figures.

Need to label the observed morphological changes in the Figure 1L.

Response: The morphological changes were labeled and explained in the legend.

Improve the quality of figure 2 plots.

Response: The figures were amended to improve the readability of the graph plots.

Which factor more dominant in the impact of GSTT2 expression on BCG immunotherapy of bladder cancer? Explain it.

Response: The first paragraph of the discussion includes a detailed explanation of the dominant impact of GSTT2 expression, which is the intracellular survival of BCG in host cells.

Authors need to cite the recent year references in the discussion part.

Response: We have updated one citation to an updated article published within the last five years. However, many articles from the 1990s cited in this manuscript are relevant as they contain the data for the observations mentioned in the discussion.

Conclusion section is missing in the present manuscript.

Response: The conclusion section is included in the updated manuscript.

To improve the manuscript, the authors could compare their research with other studies in a table to highlight the benefits of their work.

Response: We focused on comparing the clinical studies investigating the effects of genetic variances in the GST theta genes on the risk of disease development or association with therapy outcomes. The observations and comparisons are sufficient for a paragraph in the discussion. However, listing a table highlighting all the GST families is challenging for our study due to the vast literature in this field. This work is more beneficial for a review paper, with more meaningful insights and organized multi-layered content sections.

Finally, the text should be checked for grammatical, formatting, and punctuation mistakes.

Response: We have checked the article multiple times using two editing software.